

# Comparison of various approaches to tagging for the inflectional Slovak language

Lubomír Benko[1], Dasa Munkova[1], Mária Pappová[1] and Michal Munk[1,2]

[1] Department of Computer Science, Constantine the Philosopher University in Nitra, Nitra, Slovakia
[2] Science and Research Centre, University of Pardubice, Pardubice, Czech Republic

## ABSTRACT

Morphological tagging provides essential insights into grammar, structure, and the mutual relationships of words within the sentence. Tagging text in a highly inflectional language presents a challenging task due to word ambiguity. This research aims to compare six different automatic taggers for the inflectional Slovak language, seeking for the most accurate tagger for literary and non-literary texts. Our results indicate that it is useful to differentiate texts into literary and non-literary and subsequently, based on the text style to deploy a tagger. For literary texts, UDPipe2 outperformed others in seven out of nine examined tagset positions. Conversely, for non-literary texts, the RNNTagger exhibited the highest performance in eight out of nine examined tagset positions. The RNNTagger is recommended for both types of the text, the best captures the inflection of the Slovak language, but UDPipe2 demonstrates a higher accuracy for literary texts. Despite dataset size limitations, this study emphasizes the suitability of various taggers for the inflectional languages like Slovak.

Corresponding author
Lubomír Benko, lbenko@ukf.sk

## INTRODUCTION

Part-of-speech (POS) tagging is one of the most essential tasks of natural language processing (NLP), aiming to assign the correct syntactic label to each word in the context of its appearance. It is an automatic text annotation process, in which assigned words or tokens correspond to the main word class categories (adjectives, nouns, verbs, *etc.*), while they are mutually distinguished by morphosyntactic features (gender, tense, number, *etc.*). Together with lemmatization, both are fundamental tasks, and/or steps of linguistic pre-processing, which can be later used in NLP tasks such as machine translation, word sense disambiguation, question-answering analysis, *etc*. The genesis of POS tagging is based on the ambiguity of many words regarding their POS in context.

Morphology (with all its complexity) is ubiquitous among languages, which motivates researchers to design universal schemes with universal tags, such as UniMorph, or focus researchers on projects aiming at tagset universalization, such as the Universal Dependencies (UD) project or Interset for inflectional morphology, including low-resource

languages (*Kirov et al., 2018*; *Karyukin et al., 2023*) such as Slovak. The idea behind this is that a set of syntactic POS categories—universals, exists in similar form across languages, *i.e.,* they represent their cross-lingual nature (*Petrov, Das & McDonald, 2012*). The UD project provides a token-level corpus complementary to the UniMorph type-level data (*Kirov et al., 2018*).

CLARIN (Common Language Resources and Technology Infrastructure) is a digital infrastructure governed by the European Research Infrastructure Consortium, established by the European Commission in 2009. CLARIN provides access to a broad range of tools (and language data) to support research in the humanities and beyond (*Branco et al., 2023*). It offers 68 tools for part-of-speech tagging for a single language and also for multiple languages, including Slovak (Sparv, which is Språkbanken's corpus annotation pipeline infrastructure; GENIA or STEPP Tagger for annotating biomedical texts, and MorphoDiTa).

The Slovak language belongs to a family of a highly inflectional languages with complex rules for word formation and inflection. Due to the many possible word forms, classifying context for tasks like POS tagging, lemmatization, or semantic analysis is more challenging and requires larger search space and more complex classifier training (*Hladek, Stas & Juhar, 2015*). The Slovak National Corpus (SNC) (*Horák et al., 2004*) is a morphological annotated and lemmatized corpus, consisting of two sub-corpora. Both are annotated, but the smaller sub-corpus (r-mak) is annotated manually while the larger is annotated and lemmatized automatically. The tagset, designed within SNC is both positional and attributive; tags are of unequal length following inflectional paradigm, which describes the morphological (inflectional) behaviour of the word (*Garabík & Šimková, 2012*).

## Motivation

Manual assigning a POS tag to each word in text is very time and labour-consuming. This leads to the existence of various approaches and methods to automate the task, where the overall process takes a word or a sentence as input, assigns a POS tag to the word or each word in the sentence, and creates a tagged text as the output (*Hladek, Stas & Juhar, 2015*).

Only few POS tagging algorithms and tools exist which can be deployed for low-resource languages and inflectional Slavic languages. It motivated us to conduct our research, focusing on the efficiency of these algorithms and tools especially for the Slovak language, which does not only belong to above-mentioned language families, but it is also one of official European Union languages.

The aim of this article is to compare six different automatic taggers for the inflectional Slovak language. We attempt to find the best performing tagger in terms of accuracy.

There are some studies focusing on evaluation of new proposed taggers. For example, *Straka (2018)* evaluated contextualized embeddings (UDPipe2) on 54 languages in POS tagging or *Qi et al. (2020)* built on the highly accurate neural network components that enable efficient training and evaluation for more than 70 languages (Stanza). Even though a few of them focus on Slovak, they do not compare the taggers mutually and do not distinguish between literary and non-literary texts. Moreover, they evaluated tagger accuracy using the F1 measure for the entire tagset, which provides only an overall

accuracy score with gold tokenization, but lacks detailed linguistic information. In our research, we evaluated the accuracy for each position within a 15-positional tagset, enabling us to capture various grammatical aspects of the language. By categorizing the texts into literary and non-literary style, we were able to compare taggers across different text styles, a comparison that none of the studies undertook. Therefore, this study attempts to fill this gap in literature and research focusing on POS taggers for the Slovak literary and non-literary texts.

## Contribution

The theoretical contribution of our research consists in designed methodology, how to compare automatic taggers with different output formats (POS tags).

The practical contribution lies in the verification of the effectiveness of six available automatic taggers for an inflectional Slovak language.

For automatic linguistic annotation of the Slovak text we recommend to use RNNTagger and UDPipe2 regardless of text type. Both tools best capture the inflection of the Slovak language. Moreover, our results indicate that for linguistic analysis of non-literary texts, the best approach is to combine RNNTagger with UDPipe2 (the latter mentioned mainly when determining tense and negation). However, for linguistic analysis of literary texts, the best approach is also to combine these two taggers, but in reverse order, *i.e.,* only in the case of number and person we should prefer the RNNTagger over UDPipe2.

The structure of the article is as follows. The POS tagging algorithms section briefly describes principles of rule-based taggers and stochastic taggers. The Related Work section summarizes POS tagging for low-resource languages and Slavic languages. The POS taggers section describes the most known and used taggers, suitable for POS tagging of Slovak texts. The Materials and Methods section covers used dataset, selected automatic annotation tools, and applied research methodology. The Results and Discussion sections focus on the research results and their interpretations based on the performance of the taggers in terms of accuracy. The Conclusion section summarizes our findings.

## POS tagging algorithms

Currently, the process of morphological language analysis is often performed in two steps. The first step is the analysis itself, which involves assigning to each word a list of possible combinations of lemma and morphological tags. The second step is unification, where one (if possible, correct) combination of lemma-tag is selected. The analysis typically consists of selecting entries from a database of inflected word forms, followed by guessing the lemma and/or tags for words outside the dictionary. The second step is often executed using statistical methods, which require training on manually annotated corpora (*Garabík & Šimková, 2012*).

Algorithms providing POS tagging can be grouped into rule-based taggers and stochastic taggers. Rule-based taggers—require lexical knowledge—involve a large database of handwritten disambiguation rules based on the formal syntax of the given language; on the other hand, stochastic taggers—demand high computational resources—resolve tagging ambiguities using a training set to calculate the probability that a given word has

a particular tag in a specific context (*Izzi & Ferilli, 2020*). Stochastic POS taggers do not rely on syntactic analysis of the input, but on the Hidden Markov model (HMM), which captures the lexical and contextual information.

HMM is a doubly stochastic process with a basic probabilistic process that is not observable (it is hidden), but it can only be observed through another set of stochastic processes that create a sequence of observed symbols (*Rabiner & Juang, 1986*; *Jurafsky & Martin, 2020*). Information about the model's state can be obtained from the probability distribution within possible output tokens, as each state of the model creates a different distribution. The sequence of output tokens provides an overview of the state sequence in the process known as pattern theory. However, algorithms associated with HMMs are efficient for performing tasks in many real-time systems; they are often applied in speech recognition, signal processing, and in some low-level NLP tasks such as morphological annotation, information extraction from documents, or speech-to-text conversion in speech recognition (*Fink, 2008*). In HMM words are treated as the observed events (*e.g.*, words, that could be seen in the input) and hidden events (*e.g.*, their parts of speech), which can be considered as causal factors in the probabilistic model (*Blunsom, 2004*). We treat POS tags as hidden events and individual words as observed events.

HMM consists of triple parameters $\lambda = (A, B, \pi)$ defined on the set of states $Q$ and emissions $V$ (*Blunsom, 2004*). Let $Q$ be a set of states $Q = \{q_i\}_{i=1}^N$ and $V$ be a set of emissions $V = \{v_i\}_{i=1}^N$ where $\pi$ is the probability distribution function denoting the probability $\pi(q_i) = P(S_1 = q_i)$, $A$ is an $N \times N$ matrix (called the state transition matrix), where entry $a_{ij}$ is given by $a_{ij} = P(S_{t+1} = q_j | S_t = q_i)$ and $B$ is an $NxM$ matrix (called the emission matrix), where entry $b_{ij} = P(O_t = v_j | S_t = q_i)$. The parameters within HMM can be estimated from the tagged or untagged words or tokens.

### Related work

Tagging text in a highly inflectional language is a complex task due to word ambiguity, resulting in many homographs and, due to segmentation, into a set of morphemes (*Alosaimy & Atwell, 2018*). Morphological tagging offers basic information about the grammar, and/or text structure and relationships among words within the sentence. Low-resourced morphological tagging is gaining increasing recognition (*Afanasev, 2023*). The current shift to a language-independent approach for morphological disambiguation is regarded as an extension of POS tagging, jointly predicting complex morphological tags (*Toleu, Tolegen & Mussabayev, 2022*).

There are different pre-trained monolingual and multilingual models that are used for the morphological tagging, but most of them are too universal or underprepared for low resource languages, expect for the Stanza tagger (*Afanasev, 2023*). That is why we also decided to employed Stanza in our research. *Afanasev (2023)* compared Stanza to UDPipe taggers for Belarusian-Khislavichi and also for Russian-Taiga languages, all belonging to the East Slavic family, and found that a modified Stanza tagger provides more effective tagging than UDPipe. *Ljubešić & Dobrovoljc (2019)* conducted an experiment with three Slavic—Slovenian, Croatian, and Serbian—morphosyntactic taggers and compared two state-of-the-art tools with different architecture, traditional Reldi-tagger with a modified

neural Stanza (stanfordnlp+lex). They showed that the neural Stanza yields significant improvements in tagging compared to the traditional approach. *Fehle, Schmidt & Wolff (2021)* evaluated two POS-taggers for German—TreeTagger and Stanza. Concerning POS-tagging, they showed very few differences.

*Spoustová et al. (2009)* focused on evaluating the quality of morphological annotation generated by several different POS taggers. The quality assessment was conducted for the tools HMM tagger, Morče (the predecessor of the MorphoDiTa tagger), and Feature-Based Tagger. The results of the POS taggers were categorized into three cases: correct annotation, incorrect annotation, and vague annotation. The main contribution of the research was the methodology for identifying problematic tags without the need for a human-annotated baseline.

*Rosen et al. (2014)* dealt with the annotation scheme of texts produced by non-native speakers of Czech. The authors not only focused on manual annotation but also conducted experiments with automated linguistic annotation tools. The results were compared for a spell checker Korektor (*Richter, 2010*) and the POS tagger Morče, aiming to identify errors in automated tagging. Furthermore, the authors compared two taggers: Morče and TnT (*Brants, 2000*). The results demonstrated that TnT faced challenges in a context with many errors, but performed better than Morče on faulty forms. On the other hand, Morče exhibited a strong preference for verbs and demonstrated better overall performance.

*Machura et al. (2019)* compared the Czech morphological taggers MorphoDiTa and Majka Tagger. The experimental results indicated higher precision and recall for MorphoDiTa. During the experiment, the authors enhanced the MorphoDiTa tool and significantly improved its accuracy. The authors examined the differences in the Czech language. However, the conclusion of the study emphasized that the input text has a significant impact on the quality of the output.

*Straka & Straková (2017)* compared different versions of UDPipe and its subsequent enhancements. The taggers were evaluated using the TIRA platform for the CoNLL 2017 UD Shared Task, where all inputs were plain text, and the results were based on F1 scores. Overall, the system upgrades demonstrated improvements in POS tag annotation. The authors utilized the old version of UDPipe as a baseline. *Straka (2018)* continued to improve the model by refining the model architecture, resulting in enhanced performance. The evaluation was not limited to POS tagging but also included lemmatization for the tool in the CoNLL 2018 UD Shared Task. The results of the competition motivated the author to further enhance the tool in the future.

## POS taggers
### TreeTagger

TreeTagger is designed to analyze the morphological and syntactic structure of a sentence and assign parts of speech and lemmas to each word (*Schmid, 1999*). It can be used for multiple languages, including Slovak, and is adaptable to other languages if a lexicon and manually annotated corpus are available. It is primarily based on decision trees guided by modified ID3 algorithms. The tree itself is modeled recursively on a training data sample, mainly consisting of trigrams. It combines rule-based and stochastic algorithms

and uses a set of rules to identify possible parts of speech for each word in the text based on its morphological and contextual properties. These rules are then applied within a probabilistic framework to determine the most probable part of speech for each word.

## MorphoDita

The Morphological Dictionary and Tagger (MorphoDiTa) is an open-source tool for morphological text analysis of natural language. It was developed within the LINDAT project by *Straka & Straková (2014)*. It is one of the most widely used tools for morphological analysis focusing on English, Czech, and Slovak. It is based on a combination of rule-based algorithms and machine learning. Besides morphological analysis, it performs morphological generation, tagging, and tokenization. It is distributed as a standalone tool or a library, along with trained linguistic models (ibid). Its predecessors are the tagging library Morče and Featurama (*Spoustová et al., 2009*). The tagger is implemented as a supervised, averaged perceptron. It further utilizes two main machine learning algorithms:

- Morphological analyzer (Independent Feature Selection Classifier) for distinguishing various morphological features (*e.g.*, number, case, gender, *etc.*). The classifier is trained on a large source of annotated data.
- Dependency parsing for deep syntactic analysis. It identifies relationships between words/tokens in the text and creates a tree structure of dependencies.

MorphoDiTa estimates regular patterns based on affixes, common morpheme endings, and automatically groups them into morphological "templates" without language-specific knowledge. MorphoDiTa Online operates on the same principle as the library itself, available for various operation systems (Linux/Windows/OS X) and various programming languages (C++, Python, Perl, Java, and C#). The trained model is available from 2017, and extensive changes and updates have only taken place within the Czech models.

### UDPipe2

UDPipe2, similar to its predecessor UDPipe1, is a language-agnostic, trainable pipeline performing POS tagging, lemmatization, and dependency syntactic parsing using CoNLL-U format (*Straka & Straková, 2017*; *Straka, 2018*). UDPipe2 compare to UDPipe1 is a Python prototype. Trained models are available for almost all Universal Dependencies (UD) corpora. UDPipe2 utilizes multiple machine learning algorithms for morphological and syntactic analysis of texts. Specifically, the tool employs algorithms based on deep neural networks, allowing the tool to learn from data and patterns, creating more precise and efficient models for text analysis. It includes tokenizer, POS tagger, lemmatizer, and parser models for 99 treebanks of 63 languages of Universal Depenencies 2.6 Treebanks, created solely using UD 2.6 data (*Straka, 2020*).

### RNNTagger

RNNTagger was developed in 2019, primarily aiming at morphological annotation of historical texts that preserve a certain dialect with which many libraries have struggled (*Schmid, 2019*). It is a tool for annotating text with POS and lemma information. It is a type of sequence labeling model that utilizes recurrent neural networks, specifically

bidirectional long short-term memory models (Bi-LSTMs), to predict tags for each element in the sequence. It is a neural POS tagger implemented in Python using the PyTorch deep learning library. The model takes a sequence of words as input and processes them one by one while maintaining a hidden state that captures information about the context of previous words. The hidden state is updated at each time step using the current input and the previous hidden state, allowing the model to learn dependencies between words in the sequence. Compared to TreeTagger, RNNTagger lemmatizes all tokens, but requires Python and PyTorch. RNNTagger tries to lemmatize all words, including unknown words, compared to TreeTagger, which uses the word form, and RNNTagger suffers from attempting to lemmatize non-inflected tokens (*Proisl et al., 2020*).

### Stanza

Stanza is an open-source Python NLP toolkit supporting many human languages (*Qi et al., 2020*). Stanza is built on top of the PyTorch library and utilizes deep learning models in the form of neural pipelines, where each phase is implemented using a deep neural network model trained on a large amount of annotated data to perform the corresponding task. The outputs of each phase are used as inputs for the next phase, allowing the pipeline to sequentially process the input text and produce a whole range of outputs, such as syntactic and semantic representations of the text. One of the key advantages of Stanza is its ease of use, contributing to its high popularity. It provides a simple and consistent interface for performing NLP tasks, making it accessible to users with varying levels of expertise. Additionally, pre-trained models are customizable, allowing users to fine-tune them based on their data. Stanza is used in various applications, including social media analysis, machine translation, and information extraction. Compared to UDPipe, Stanza supports 66 languages and is fully neural.

Table 1 contains the summarization of the examined automatic POS taggers that support the automatic annotation of Slovak language.

## MATERIALS & METHODS

### POS categories

The part of speech category (POS) is fundamental, as each morphological interpretation of a word form is assigned a POS value (*Petkevič et al., 2019*). Tagsets for different languages are typically different. They can be entirely different for unrelated languages and very similar for related languages, but this is not always the rule. Tagsets can also vary in levels of granularity. Basic tagsets may contain tags for the most common parts of speech (N for noun, V for verb, A for adjective, *etc.*) (*Universal Dependencies contributors, 2022*). However, it is more common to go into detail and differentiate between singular and plural nouns, verb phrases, tenses, aspects, voices, and more. *Petrov, Das & McDonald (2012)* proposed a tagset consisting of twelve universal POS categories for 22 different languages, including Czech, but not Slovak: NOUN (nouns), VERB (verbs), ADJ (adjectives), ADV, (adverbs), PRON (pronouns), DET (determiners and articles), ADP (prepositions and postpositions), NUM (numerals), CONJ (conjunctions), PRT (particles), '.' (punctuation marks) and X (abbreviations or foreign words). Hajic developed the Czech Prague

**Table 1 Comparison of examined automatic POS taggers.**

| Automatic POS Tagger | Supported Languages | Study |
| --- | --- | --- |
| MorphoDiTa | 3 (EN, CZ, SK) | *Straka & Straková (2014)* |
| MorphoDiTa _Online | 3 (EN, CZ, SK) | *Straka & Straková (2014)* |
| Stanza | 66 (incl. EN, DE, SK) | *Qi et al. (2020)* |
| UDPipe2 | 63 (incl. EN, DE, SK) | *Straka & Straková (2017)*, *Straka (2018)* and *Straka (2020)* |
| TreeTagger | 35 (incl. EN, DE, SK) | *Schmid (1999)* |
| RNNTagger | 35 (incl. EN, DE, SK) | *Schmid (2019)* |

Dependency Treebank with its tagset format (PDT tagset), designed for the needs of Slavic languages (*Hajič, 2006*; *Bejček & Straňák, 2010*). The PDT tagset uses three layers of annotation—morphological, syntactical, and tectogrammatical. It is a string of 15 characters that can more precisely determine the meaning of the tagged word; one character symbol encodes one morphological category (*Hajič et al., 2020*). The PDT tagset consists of a fixed length, and each position encodes one grammatical category, while two positions (13th and 14th) are empty. The attributes in positional tags are as follows: the 1st position—part of speech, 2nd—a detailed part of speech, 3rd—gender, 4th—number, 5th—case, 6th—possessor's gender, 7th—possessor's number, 8th—person, 9th—tense, 10th—degree of comparison, 11th—negation, 12th—voice, 13th—empty, 14th—empty, and the last position, 15th—variant and style.

According to general principles, two main groups of tagsets are recognized—UPOS and XPOS. The Universal POS tag (UPOS) is represented by tags indicating the basic categories of parts of speech. Universal POS tags are categorized into open class words, closed class words, and others. Under open class words, words can be continually added and modified. Over the decades, words like "smartphone", "selfie", or "e-sport" have been added to nouns. On the other hand, closed class words represents a group of words that will be universally valid and immutable (*Universal Dependencies contributors, 2022*). Under the others category, various symbols, punctuation marks, or symbols for unrecognizable words that, for example, the model cannot identify, are recognized.

The abbreviation XPOS refers to a language-specific POS tag specific to a given language (*e.g.*, English: Language-specific POS). Unique rules for encoding XPOS are defined by each library. A single character, which can be a letter of the Latin alphabet, a digit, or a mathematical symbol, represents the values of individual categories. Consequently, each letter corresponds to only one value, even across parts of speech. An exception is observed in the paradigm category, which reuses the part of speech code (*Garábik & Bobeková, 2021*). One tag (label) for one token and lemma is formed by a set of these characters.

## Dataset

Obtaining a dataset that both, best captures the diversity of the Slovak language and is sufficiently complex for evaluating tools specialized in the Slovak language tagging was challenging. Manually annotated data are crucial for training and evaluating statistical tools such as POS taggers and lemmatizers (*Proisl et al., 2020*). For this purpose, a manually

annotated and lemmatized sub-corpus of the Slovak Dependency Corpus (SDC) was chosen (*Gajdošová & Šimková, 2016*). The dataset consists of 10,604 sentences and 106,043 tokens. The annotation follows the guidelines of the Prague Dependency Treebank (PDT) (*Hajič, 2006*) slightly modified to align with Slovak grammatical rules. Morphological tags, lemmas, and dependencies were manually assigned to each word. The sub-corpus includes only sentences in which two human annotators perfectly agreed on the tag. A drawback of the dataset is that it mainly contains short sentences (*Benko & Benková, 2022*). The dataset also only includes surface-dependent (analytical) trees and does not encompass a deep syntactic/semantic (tectogrammatical) layer (*Majchráková et al., 2014*). The primary starting point for the Slovak annotation is the functional-generative approach. Unlike the PDT, which exclusively contains journalistic texts, the stylistic-genre structure of SDC is more diverse (*Šimková & Gajdošová, 2008*). The texts were divided into two text types: non-literary and literary. Literary texts consist of novels and fairy tales. Non-literary texts consist of historical texts, texts obtained from Wikipedia, and journalistic texts.

The data file is available in a specific dictionary-like format (CoNLL-X format) used for text processing and morphological annotation. In the CoNLL-X format, each word in a sentence is represented as a row with various columns of information, including the word form, part of speech, lemma, and syntactic head. Every dictionary created for machine learning and deep learning is stored in this format.

A more detailed description of individual columns can be found in *Gajdošová & Šimková (2016)*. For research purposes, only "POSTAG" column was investigated, while it contains manually annotated words from the "FORM" column. This type of data cannot be analyzed using the automatic tools, so it was necessary to reconstruct these data to the original format. The individual words were extracted from the file and reconstructed into sentence structures, which were then implemented as input for the examined automatic taggers.

## Taggers

Open-source libraries or tools for morphological tagging were chosen as taggers for the Slovak language. However, upon closer examination, it was revealed that another library is already utilized by a given library or tool to process NLP for the Slovak language. Among such online tools, Sparv by Swedish developers (*Hammarstedt et al., 2022*) can be mentioned; it uses the Stanza tagging library for morphological annotation of Slovak. Their priority was primarily the analysis of the Swedish Språkbanken corpus. Another library is the GENIA Tagger by *Tsuruoka et al. (2005)*, which annotates the Slovak language, but is only intended for biomedical texts, as is the STEPP Tagger (*Piao, Tsuruoka & Ananiadou, 2009*), which originated at the University of Tokyo and was further presented in 2012. Lastly, the Turku Neural Parse Pipeline by Finnish authors (*Kanerva et al., 2018*) is mentioned, which offers morphological annotation for over 50 languages, but the tagset for the Slovak language was not available. Finally, a library is mentioned that morphologically annotates the Slovak language, but its implementation was unsuccessful. The first such library was Dagger, which originated at the Technical University of Košice and was based on the principle of HMM, specifically the Viterbi algorithm and binary decision trees, particularly

the ID3 algorithm (*Hládek, Staš & Juhár, 2012*). This library could not be implemented because it was no longer supported by newer versions of the Linux operating system. Another problematic tool was the RFTagger (*Schmid & Laws, 2008*), based on the HMM and decision trees method. It is suitable mainly for POS tagsets with many fine-grained tags, *i.e.,* those that contain more details and distinguish between various subtypes of parts-of-speech and grammatical categories. Despite the excellent results achieved by this library on the German Tiger Treebank, it was not possible to annotate the entire text correctly, leading to the exclusion of RFTagger from further evaluation. For this reason, the following automated taggers were selected for comparison: TreeTagger, RNNTagger, Stanza, MorphoDiTa (application and online version will be separated), and UDPipe2.

## Applied methodology

The applied methodology, inspired by other research (*Hochreiter & Schmidhuber, 1997*; *Huang, Xu & Yu, 2015*; *Yao & Huang, 2016*; *Benkova et al., 2021*; *Munkova et al., 2021a*; *Munkova et al., 2021b*; *Kapusta et al., 2021*), comprises the following steps (Fig. 1):

(1) Acquisition of dataset with manual morphological annotation—a gold tokenization (*Gajdošová & Šimková, 2016*). This gold tokenization is positional and attributive, and the tags are of unequal length following inflectional paradigm. It contains 85,929 tokens.

(2) Data preparation—sentence reconstruction. Since the acquired dataset was already tokenized, we converted the tokenized texts into the original text forms, *i.e.,* to sentences, giving us 10,604 sentences (85,929 tokens). Afterwards, we divided texts according to the text type into literary texts (45,819 tokens) and non-literary texts (40,110 tokens).

(3) Automatic POS tagging—converted texts (literary and non-literary texts) were annotated using the following tools:

 (a) RNNTagger (using the model, trained on the SNC),

 (b) TreeTagger (using the model Slovak parameter file, trained on the SNC),

 (c) MorphoDiTa (using the model slovak-morfflex-pdt-170914 (*Hajič & Hric, 2017*), trained on SNC),

 (d) MorphoDiTa online (using the model slovak-morfflex-pdt-170914 (*Hajič & Hric, 2017*)),

 (e) Stanza (we used the model from Universal Dependencies v2.12 (*Zeman et al., 2023*), trained on SNC),

 (f) UDPipe2 (we used the model slovak-snk-ud−2.12-230717, trained on SNC). The tools were run at two different personal computers because RNNTagger only requires the Linux operating system. RNNTagger was implemented on ASUS Transformer Book Flip TP300LD (Intel Core i5 4210 Haswell, RAM 12GB DDR3L, NVIDIA GeForce GT 820 2GB, 500 GB SSD, Ubuntu Desktop 22.10). Other tagging tools where implemented on Apple M1 Max (Processor with 10 cores 2.06–3.22 GHz, 32-Core GPU, 16-Core Neural Engine, 32 GB RAM, 1 TB SSD, macOS Ventura 13.2.1).

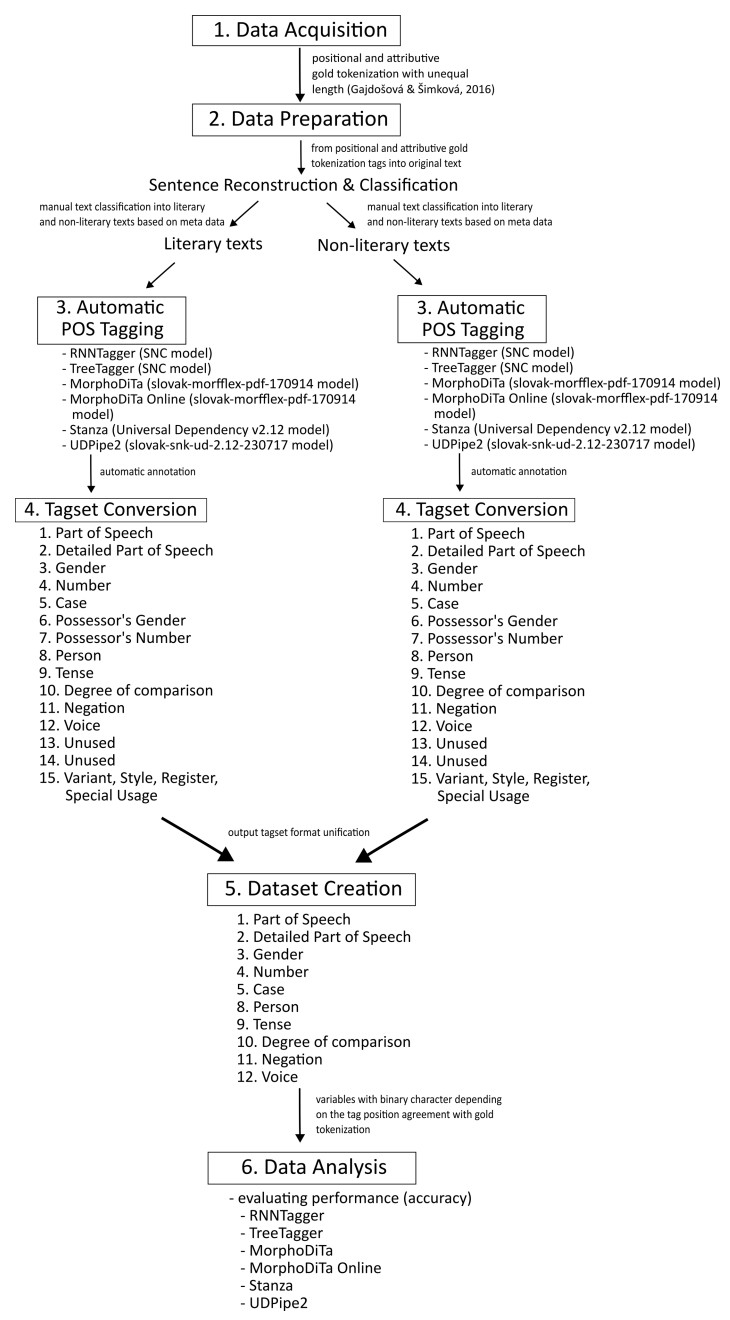

**Figure 1   Methodology diagram.**

(4)  Tagset conversion—the automatic taggers (RNNTagger, TreeTagger, UDPipe2, and Stanza) do not employ the same PDT output tagset format, so their output tagsets need to be converted to the same tagset format. To compare the performance of the investigated taggers in terms of accuracy with the gold tokenization (reference), we unified the output tagset format. Since the tagset used for Slovak has tags of unequal

length, *i.e.,* a different number of characters (positions) for each part of speech, we decided to employ a universal, 15-positional tagset for all examined taggers.

(5) Dataset creation—the outputs of the six automated taggers were joint into one data matrix and dummy variables were created for each position (1st–15th). These variables have a binary character, depending on the tag position match (agreement) with the reference (gold tokenization). Some positions did not contain any tags, so these positions were excluded from the experiment. We created two datasets depending on text type (literary and non-literary dataset).

(6) Data analysis—we applied non-parametric procedures based on both, frequency (Cochran Q test) and ranks, where the degree of concordance was expressed by the Kendall coefficient of concordance.

## RESULTS

As described in the methodology, some of the automatic POS taggers did not support the PDT tagset style. During the conversion into the PDT tagset style some empty positions were generated, which resulted in the elimination of the positions 6th, 7th, 13th, 14th and 15th from the experiment. Positions 13th and 14th are empty in general, and the position 6th and 7th within a tagset are associated with the possessor's gender and the possessor's number, and the position 15th with the style.

Since the tagsets, produced by the six examined automatic taggers, representing the indicators of accuracy (agreement with a reference tagset), have a binary character at the relevant tag position, we used non-parametric procedures. Based on frequencies (Cochran Q test) and ranks (Kendall coefficient of concordance), we tested the global null statistical hypotheses, which claim that there is no statistically significant difference in the performance of the investigated taggers in terms of accuracy with reference. We tested the hypothesis for each tag of the 15-positional tagset, except for the five excluded tags (6th, 7th, 13th, 14th, and 15th). We examined the performance of the automatic taggers separately for literary and non-literary texts, to determine whether the text type can affect the performance of given tagger.

### Literary and non-literary texts

In the case of non-literary texts, based on the results of Cochran Q tests, the global null hypothesis (stating there are no statistically significant differences between individual tags within the tagset produced by examined POS taggers and a reference tags in tagset) is rejected at the 0.001 significance level for tags in the 1st, 2nd, 3rd, 4th, 5th, 8th, 9th, 10th, 11th, and 12th position in the tagset (1st: $N = 45819$, $Q = 18024.87$, $df = 5$, $p < 0.001$; 2nd: $N = 38967$, $Q = 34381.00$, $df = 5$, $p < 0.001$; 3rd: $N = 31285$, $Q = 24833.86$, $df = 5$, $p < 0.001$; 4th: $N = 34125$, $Q = 16198.12$, $df = 5$, $p < 0.001$; 5th: $N = 32531$, $Q = 12497.83$, $df = 5$, $p < 0.001$; 8th: $N = 6672$, $Q = 19756.04$, $df = 5$, $p < 0.001$; 9th: $N = 6169$, $Q = 535.99$, $df = 5$, $p < 0.001$; 10th: $N = 7223$, $Q = 4400.26$, $df = 5$, $p < 0.001$; 11th: $N = 7377$, $Q = 1078.99$, $df = 5$, $p < 0.001$; 12th: $N = 7210$, $Q = 15876.04$, $df = 5$, $p < 0.001$).

**Table 2  Ranking of taggers in the 1st tagset position (a) non-literary texts, (b) literary texts.**

| (a) $N = 45819$ | 1's (%) | 1 | 2 | 3 | 4 | 5 | 6 |
|---|---|---|---|---|---|---|---|
| **MorphoDiTa** | 86.65 | o | | | | | |
| **MorphoDiTa _Online** | 91.10 | | o | | | | |
| Stanza | 97.51 | | | o | | | |
| UDPipe2 | 98.10 | | | | o | | |
| TreeTagger | 98.71 | | | | | o | |
| RNNTagger | 99.37 | | | | | | o |

| (b) $N = 40110$ | 1's (%) | 1 | 2 | 3 | 4 | 5 |
|---|---|---|---|---|---|---|
| **MorphoDiTa** | 85.68 | | o | | | |
| **MorphoDiTa _Online** | 89.08 | | | o | | |
| TreeTagger | 98.44 | | | | o | |
| Stanza | 98.80 | | | | | o |
| RNNTagger | 99.21 | o | | | | |
| UDPipe2 | 99.26 | o | | | | |

**Notes.**
o, Homogenous Groups, $p > 0.05$, marked - similar tagger ranking for both text styles.

Similarly, for the literary texts, based on the results of Cochran Q tests, the global null hypotheses is rejected at the 0.001 significance level for tags in the 1st, 2nd, 3rd, 4th, 5th, 8th, 9th, 10th, 11th, and 12th position (1st: $N = 40110$, $Q = 20352.07$, $df = 5$, $p < 0.001$; 2nd: $N = 28956$, $Q = 40532.13$, $df = 5$, $p < 0.001$; 3rd: $N = 24871$, $Q = 38023.70$, $df = 5$, $p < 0.001$; 4th: $N = 28952$, $Q = 18599.09$, $df = 5$, $p < 0.001$; 5th: $N = 21590$, $Q = 11327.92$, $df = 5$, $p < 0.001$; 8th: $N = 11077$, $Q = 36999.44$, $df = 5$, $p < 0.001$; 9th: $N = 9714$, $Q = 303.93$, $df = 5$, $p < 0.001$; 10th: $N = 4515$, $Q = 7028.828$, $df = 5$, $p < 0.001$; 11th: $N = 11779$, $Q = 4927.626$, $df = 5$, $p < 0.001$; 12th: $N = 11643$, $Q = 30379.50$, $df = 5$, $p < 0.001$).

Our results indicate that there are differences in the accuracy of taggers' performance, whether the text is literary or non-literary. Moreover, these differences are statistically significant.

Non-parametric procedures, which we applied, work with absolute differences of sums of ranks, where critical values were obtained asymptotically. Based on the multiple comparisons, we identified homogeneous groups among which the statistically significant differences (Tables 2, 3, 4, 5, 6, 7, 8, 9, 10 and 11) were proven.

## The 1st tagset position—part of speech

In addition to the ten traditional parts of speech—noun (N), adjective (A), pronoun (P), verb (V), adverb (D), numeral (C), conjunction (J), preposition (R), interjection (I), and particle (T)—the 1st tagset position distinguishes also the abbreviation (B), foreign word (F), segment (S), isolated letter (Q), and punctuation (Z).

For both types of texts, MorphoDiTa Tagger, with less than 87% (86.65% for non-literary and 85.68% for literary texts), achieved the lowest performance in terms of matching with the reference (Table 2). On the other hand, the RNNTagger, with more than 99%, achieved the highest performance in terms of matching with the reference in the case of non-literary

**Table 3 Ranking of taggers in the 2nd position of the tag (a) non-literary texts, (b) literary texts.**

| (a) N = 38967 | 1's (%) | 1 | 2 | 3 | 4 | 5 |
|---|---|---|---|---|---|---|
| **MorphoDiTa** | 75.69 | | o | | | |
| **MorphoDiTa_Online** | 80.23 | | | o | | |
| Stanza | 96.71 | | | | o | |
| UDPipe2 | 98.30 | o | | | | |
| TreeTagger | 98.52 | o | | | | |
| RNNTagger | 99.18 | | | | | o |

| (b) N = 28956 | 1's (%) | 1 | 2 | 3 | 4 |
|---|---|---|---|---|---|
| **MorphoDiTa** | 66.30 | | | o | |
| **MorphoDiTa_Online** | 70.01 | | | | o |
| TreeTagger | 97.82 | o | | | |
| Stanza | 97.95 | o | | | |
| RNNTagger | 98.83 | | o | | |
| UDPipe2 | 99.13 | | o | | |

Notes.
o, Homogenous Groups, $p > 0.05$, marked - similar tagger ranking for both text styles.

**Table 4 Ranking of taggers in the 3rd position of the tag (a) non-literary texts, (b) literary texts.**

| (a) N = 31285 | 1's (%) | 1 | 2 | 3 | 4 | 5 |
|---|---|---|---|---|---|---|
| **MorphoDiTa** | 74.59 | | o | | | |
| **MorphoDiTa_Online** | 79.59 | | | o | | |
| Stanza | 95.24 | o | | | | |
| UDPipe2 | 95.59 | o | | | | |
| TreeTagger | 98.19 | | | | o | |
| RNNTagger | 98.67 | | | | | o |

| (b) N = 24871 | 1's (%) | 1 | 2 | 3 | 4 |
|---|---|---|---|---|---|
| **MorphoDiTa** | 63.17 | | | o | |
| **MorphoDiTa_Online** | 67.64 | | | | o |
| TreeTagger | 97.78 | | o | | |
| Stanza | 98.21 | o | o | | |
| RNNTagger | 98.56 | o | | | |
| UDPipe2 | 98.65 | o | | | |

Notes.
o, Homogenous Groups, $p > 0.05$, marked - similar tagger ranking for both text styles.

texts, and in the case of literary texts, UDPipe2 (99.26%) achieved the highest accuracy and/or performance with the reference (Table 2b).

For non-literary texts (Table 2a), six trivial single-element homogenous groups were identified among which statistically significant differences were observed ($p <0.05$).

For literary texts (Table 2b), four trivial single-element homogenous groups and one two-element homogenous group were identified among which statistically significant differences were observed ($p < 0.05$). In terms of agreement with the reference, the RNNTagger and UDPipe2 form one two-element homogenous group ($p > 0.05$).

**Table 5  Ranking of taggers in the 4th position of the tag (a) non-literary texts, (b) literary texts.**

| (a) $N = 34125$ | 1's (%) | 1 | 2 | 3 | 4 |
|---|---|---|---|---|---|
| **MorphoDiTa** | 85.13 | | | o | |
| **MorphoDiTa_Online** | 90.69 | | | | o |
| Stanza | 97.97 | o | | | |
| UDPipe2 | 98.24 | o | | | |
| TreeTagger | 99.24 | | o | | |
| **RNNTagger** | 99.51 | | o | | |

| (b) $N = 28952$ | 1's (%) | 1 | 2 | 3 |
|---|---|---|---|---|
| **MorphoDiTa** | 83.60 | | o | |
| **MorphoDiTa_Online** | 88.10 | | | o |
| TreeTagger | 99.33 | o | | |
| Stanza | 99.47 | o | | |
| UDPipe2 | 99.55 | o | | |
| **RNNTagger** | 99.62 | o | | |

**Notes.**
o, Homogenous Groups, $p > 0.05$, marked - similar tagger ranking for both text styles.

**Table 6  Ranking of taggers in the 5th position of the tag (a) non-literary texts, (b) literary texts.**

| (a) $N = 32531$ | 1's (%) | 1 | 2 | 3 | 4 | 5 | 6 |
|---|---|---|---|---|---|---|---|
| **MorphoDiTa** | 83.79 | o | | | | | |
| **MorphoDiTa_Online** | 88.95 | | o | | | | |
| Stanza | 95.58 | | | o | | | |
| UDPipe2 | 96.77 | | | | o | | |
| TreeTagger | 97.76 | | | | | o | |
| RNNTagger | 98.72 | | | | | | o |

| (b) $N = 21590$ | 1's (%) | 1 | 2 | 3 | 4 |
|---|---|---|---|---|---|
| **MorphoDiTa** | 83.65 | | o | | |
| **MorphoDiTa_Online** | 88.75 | | | o | |
| TreeTagger | 97.72 | | | | o |
| Stanza | 98.81 | o | | | |
| RNNTagger | 98.86 | o | | | |
| UDPipe2 | 99.09 | o | | | |

**Notes.**
o, Homogenous Groups, $p > 0.05$, marked - similar tagger ranking for both text styles.

The taggers' performance for both text types is higher than 97% (except for MorphoDiTa and MorphoDiTa_online), which indicates that the part of speech identification (1st position in the PDT tagset format) is mostly accurate, even though the best performance was obtained from the RNNTagger for both text types (Table 2).

**Table 7  Ranking of taggers in the 8th position of the tag (a) non-literary texts, (b) literary texts.**

| (a) N = 6672 | 1's (%) | 1 | 2 |
|---|---|---|---|
| **MorphoDiTa** | 39.40 | | o |
| **MorphoDiTa_Online** | 39.51 | | o |
| Stanza | 99.10 | o | |
| UDPipe2 | 99.21 | o | |
| TreeTagger | 99.43 | o | |
| **RNNTagger** | 99.70 | o | |

| (b) N = 11077 | 1's (%) | 1 | 2 | 3 |
|---|---|---|---|---|
| **MorphoDiTa** | 31.23 | | o | |
| **MorphoDiTa_Online** | 31.26 | | o | |
| TreeTagger | 98.28 | | | o |
| UDPipe2 | 99.03 | o | | |
| Stanza | 99.22 | o | | |
| **RNNTagger** | 99.55 | o | | |

Notes.
o, Homogenous Groups, $p > 0.05$, marked - similar tagger ranking for both text styles.

**Table 8  Ranking of taggers in the 9th position of the tag (a) non-literary texts, (b) literary texts.**

| (a) N = 6169 | 1's (%) | 1 | 2 |
|---|---|---|---|
| MorphoDiTa | 98.02 | | o |
| MorphoDiTa_Online | 98.02 | | o |
| TreeTagger | 99.82 | o | |
| RNNTagger | 99.82 | o | |
| **UDPipe2** | 99.90 | o | |
| **Stanza** | 99.94 | o | |

| (b) N = 11077 | 1's (%) | 1 | 2 |
|---|---|---|---|
| MorphoDiTa_Online | 99.24 | | o |
| MorphoDiTa | 99.24 | | o |
| RNNTagger | 99.89 | o | |
| TreeTagger | 99.90 | o | |
| **UDPipe2** | 99.94 | o | |
| **Stanza** | 99.94 | o | |

Notes.
o, Homogenous Groups, $p > 0.05$, marked - similar tagger ranking for both text styles.

## The 2nd tagset position—a detailed part of speech

The second tagset position contains values for fine-grained distinction of the major POS category (66 SUBPOS values) which serves as an indicator of applicability/non-applicability of other categories (*Mikulová et al., 2020*).

Ranking of taggers (Table 3) has again shown that the lowest performance in terms of matching with the reference in the 2nd position was achieved by the MorphoDiTa tagger (75.69% for non-literary texts and 66.30% for literary texts). Similar to the 1st position, the RNNTagger, with more than 99%, has achieved the highest performance in terms

**Table 9   Ranking of taggers in the 10th position of the tag (a) non-literary, (b) literary texts.**

| (a) $N = 7223$ | 1's (%) | 1 | 2 | 3 | 4 | 5 |
|---|---|---|---|---|---|---|
| **MorphoDiTa** | 80.42 | | o | | | |
| **MorphoDiTa_Online** | 83.65 | | | o | | |
| **Stanza** | 96.01 | | | | o | |
| UDPipe2 | 97.73 | o | | | | |
| TreeTagger | 98.10 | o | | | | |
| RNNTagger | 98.88 | | | | | o |

| (b) $N = 4515$ | 1's (%) | 1 | 2 | 3 | 4 |
|---|---|---|---|---|---|
| **MorphoDiTa** | 62.97 | | o | | |
| **MorphoDiTa_Online** | 64.89 | | | o | |
| **Stanza** | 97.56 | o | | | |
| TreeTagger | 97.72 | o | | | |
| RNNTagger | 97.96 | o | | | |
| UDPipe2 | 99.56 | | | | o |

**Notes.**
o, Homogenous Groups, $p > 0.05$, marked - similar tagger ranking for both text styles.

**Table 10   Ranking of taggers in the 11th position of the tag (a) non-literary, (b) literary texts.**

| (a) $N = 7377$ | 1's (%) | 1 | 2 | 3 |
|---|---|---|---|---|
| **MorphoDiTa** | 94.89 | | o | |
| **MorphoDiTa_Online** | 95.58 | | | o |
| **TreeTagger** | 98.98 | o | | |
| **RNNTagger** | 99.08 | o | | |
| **Stanza** | 99.27 | o | | |
| **UDPipe2** | 99.32 | o | | |

| (b) $N = 11779$ | 1's (%) | 1 | 2 | 3 |
|---|---|---|---|---|
| **MorphoDiTa** | 90.14 | | o | |
| **MorphoDiTa_Online** | 90.98 | | | o |
| **TreeTagger** | 99.38 | o | | |
| **RNNTagger** | 99.58 | o | | |
| **Stanza** | 99.64 | o | | |
| **UDPipe2** | 99.69 | o | | |

**Notes.**
o, Homogenous Groups, $p > 0.05$, marked - similar tagger ranking for both text styles.

of matching with the reference for non-literary texts (Table 3a), and UDPipe2 (99.13%) achieved the highest performance and/or accuracy with the reference for literary texts (Table 3b).

For non-literary texts (Table 3a) five homogenous groups were identified among which statistically significant differences were observed ($p < 0.05$)—four single-element homogenous groups and one two-element homogenous group, which consists of UDPipe2 and TreeTagger.

**Table 11  Ranking of taggers in the 12th position of the tag (a) non-literary texts, (b) literary texts.**

| (a) N = 7210 | 1's (%) | 1 | 2 |
|---|---|---|---|
| **MorphoDiTa** | 53.37 | | o |
| **MorphoDiTa_Online** | 53.47 | | o |
| UDPipe2 | 98.53 | o | |
| **Stanza** | 98.59 | o | |
| TreeTagger | 99.24 | o | |
| RNNTagger | 99.29 | o | |

| (b) N = 11643 | 1's (%) | 1 | 2 |
|---|---|---|---|
| **MorphoDiTa** | 46.58 | | o |
| **MorphoDiTa_Online** | 46.71 | | o |
| **Stanza** | 99.51 | o | |
| RNNTagger | 99.54 | o | |
| UDPipe2 | 99.54 | o | |

**Notes.**

o, Homogenous Groups, $p > 0.05$, marked - similar tagger ranking for both text styles.

For literary texts (Table 3b) four homogenous groups (two single-element and two two-element homogenous groups) were identified among which statistically significant differences were observed ($p < 0.05$). The TreeTagger and Stanza (with more than 97% of concordance) form a one two-element homogenous group ($p > 0.05$), and the RNNTagger and UDPipe2, similarly to the first position, form the second two-element homogenous group ($p > 0.05$) in terms of agreement with the reference.

The performance of the four taggers (Stanza, UDPipe2, TreeTagger, and RNNTagger) was above 96% which indicates that the identification of the detailed part-of-speech (the 2nd tagset position) is very accurate. Moreover, we can observe similar performance for above-mentioned taggers as for the first tagset position, which confirms the link (relationship) between the first and second position.

### The 3rd tagset position—gender

The third tagset position denotes grammatical gender for both, lexical gender of nouns and agreement gender of verbs, adjectives, pronouns, and numerals (*Mikulová et al., 2020*).

Similar results of the taggers ranking for non-literary texts in the 3rd tagset position (Table 4a) have been obtained as for the previous positions. The lowest performance in terms of matching with the reference was achieved by the MorphoDiTa Tagger, with less than 75%, and the highest performance was achieved by the RNNTagger, with more than 98%. The second-highest performance, with more than 98%, was achieved by TreeTagger, which is a predecessor of the RNNTagger.

Five homogenous groups (four single-element homogenous groups and one two-element homogenous group) were identified (Table 4a), and statistically significant differences among them were observed in terms of agreement with the reference in the 3rd position ($p < 0.05$).

In the case of determining the tag in the 3rd position within the literary texts (Table 4b), MorphoDiTa Tagger (with less than 64%) statistically significantly performed the worst.

Four homogenous groups were identified, but between homogenous group consisting of TreeTagger and Stanza, and homogenous group containing Stanza, RNNTagger, and UDPipe2, a statistically significant difference was not proven, only between TreeTagger and RNNTagger, and/or TreeTagger and UDPipe2.

Similarly, as for the 2nd position, the taggers TreeTagger, Stanza, RNNTagger, and UDPipe2 achieved the highest performance for both text styles (more than 95%). The results indicate that the RNNTagger is a suitable automatic tool for grammatical gender identification regardless of text type.

### The 4th tagset position—number

The fourth tagset position has mostly two standard values—singular and plural, which are also applied to adjectives, pronouns, and numerals (*Mikulová et al., 2020*).

The ranking of taggers for both text types in the 4 h position (Table 5) has shown the lowest performance in terms of matching with the reference for the MorphoDiTa Tagger, with less than 86% (85.13% for non-literary texts and 83.60% for literary texts). The highest performance in terms of matching with the reference was achieved by the RNNTagger with more than 99%.

Four homogenous groups were identified for non-literary texts (Table 5a) and three homogenous groups were identified for non-literary texts (Table 5b). Statistically significant differences were observed in terms of agreement with the reference for all the taggers determining the 4th position ($p < 0.05$). The highest agreement was achieved for the homogenous group—Stanza, RNNTagger, TreeTagger, and UDPipe2—with more than 99% in terms of matching with the reference ($p > 0.05$).

The results (Table 5) show that for both text types, the best performance was achieved by RNNTagger (>99.5%). MorphoDiTa and MorphoDiTa_Online achieved the lowest performance, but still with more than 83% matches with reference for both text types (a little higher performance was achieved for non-literary texts).

### The 5th tagset position—case

Slovak usually distinguishes among six (seven) cases: nominative, genitive, dative, accusative, (vocative), locative, and instrumental.

For non-literary texts, the results of the taggers' ranking in the 5th position copies the previous positions, above all the first tagset position (Table 6a). The lowest performance in terms of matching with the reference was achieved by the MorphoDiTa Tagger (less than 84%). The highest performance, more than 98%, was achieved by the RNNTagger. Six trivial single-element homogenous groups were identified (Table 6a). A statistically significant differences were observed in terms of agreement with the reference for all examined taggers determining the 5th tagset position ($p < 0.05$).

For literary texts, we obtained different results compared to non-literary texts (Table 6b). The worst and statistically significant performance was achieved by the MorphoDiTa Tagger (less than 84%). The highest performance was achieved by the RNNTagger, which forms together with Stanza and UDPipe2 one homogenous group with more than 98% of matching with the reference (Table 6b). Overall, four homogenous groups were identified

among which statistically significant differences were observed in terms of agreement with the reference ($p < 0.05$).

The results (Table 6) for determining the 5th tagset position show a good performance from almost all taggers. Regardless of text type, the RNNTagger proves to be one of the most suitable automatic tools for POS tagging (achieving more than 98.7%).

### The 8th tagset position—person

The eighth tagset position expresses the person of verb forms (if applicable) and person of personal pronouns; and it usually takes on three standard values—1st person, 2nd person, and 3rd person (*Mikulová et al., 2020*).

For non-literary texts (Table 7a), taggers' ranking determining the 8th position has shown two homogenous groups ($p < 0.05$)—one containing MorphoDiTa and MorphoDiTa_Online, and the second homogenous group consisting of the remaining taggers—Stanza, RNNTagger, TreeTagger, and UDPipe2. The first taggers' homogenous group achieved the lowest performance in terms of matching with the reference (less than 40%). On the other hand, the second taggers' homogenous group achieved a performance of more than 99% in terms of matching with the reference.

For literary texts (Table 7b), the worst and statistically significant performance was again achieved by the MorphoDiTa and MorphoDiTa_Online Tagger (less than 32%) forming one homogenous group ($p > 0.05$). Similarly, to non-literary texts, the highest performance was achieved by UDPipe2, Stanza, and RNNTagger (more than 99%) which form one homogenous group ($p > 0.05$).

The results (Table 7) indicate that MorphoDiTa and MorphoDiTa_Online lag behind in determining the person of verb forms or in person of personal pronouns compared to other taggers (less than 40% for non-literary texts and less than 32% for literary texts).

### The 9th tagset position—tense

The ninth tagset position represents only verb forms, in the purely morphological sense—future, present, and past (*Mikulová et al., 2020*).

Taggers' ranking determining the 9th position for non-literary texts (Table 8a) and also for literary texts (Table 8b) has shown two homogenous groups, and a statistically significant difference was observed in terms of agreement with the reference in the 9th position ($p < 0.05$). There was a small difference in performance between the two identified homogenous groups. The lowest performance in terms of matching with the reference was achieved by the MorphoDiTa and MorphoDiTa_online Tagger (about 98% for non-literary texts and about 99% for literary texts). The highest performance was obtained by the second homogenous group consisting of TreeTagger, RNNTagger, UDPipe2, and Stanza (more than 99.8%).

When determining tense, but in the purely morphological sense (9th position), Stanza proves to be the most effective tool with respect to the reference and regardless of text type.

### The 10th tagset position—degree of comparison

The tenth tagset position is used for adjective and adverbs—positive, comparative, and superlative, apart from possessive adjectives.

Taggers' ranking determining the 10th position for non-literary texts has shown five homogenous groups—four single-element and one two-element homogenous groups (Table 9a), and the statistically significant differences were observed among them in terms of agreement with the reference in the 10th position ($p < 0.05$). The lowest performance in terms of agreement with the reference was achieved by the MorphoDiTa Tagger (less than 81%). The highest performance was identified by the RNNTagger (more than 98%). Between UDPipe2 and TreeTagger was not identified a statistically significant difference in terms of agreement with the reference ($p > 0.05$).

For literary texts taggers' ranking determining the 10th position (Table 9b) has shown statistically significant differences among four homogenous groups ($p < 0.05$). A homogenous group formed by Stanza, TreeTagger, and RNNTagger achieved a high performance of more than 97%, but a statistically significant difference among them was not observed in terms of agreement with the reference ($p > 0.05$). The worst performance was achieved for the MorphoDiTa Tagger (less than 63%) followed by MorphoDiTa_online (less than 65%). Compared to the non-literary texts, in which the highest performance was achieved by the UDPipe2 Tagger with more than 99%.

When determining the tenth position (Table 9), the combination of UDPipe2 and RNNTagger is shown to be the most effective way of automatically determining the degree of comparison with respect to the reference, regardless of the type of text.

### The 11th tagset position—negation

The eleventh tagset position is fully inflectional category, as the negation in Slovak is expressed by a prefix—affirmative or negated. Negation belongs to verbs, adverbs, adjectives, and nouns (*Mikulová et al., 2020*).

Ranking of taggers for non-literary and also literary texts, determining the 11th position (Table 10), has shown a statistically significant difference in terms of agreement with the reference in the 11th position among three homogenous groups ($p < 0.05$). First two homogenous groups are single-element, and the third homogenous groups is formed by four taggers—TreeTagger, RNNTagger, Stanza, and UDPipe2 (Table 10). The lowest performance in terms of matching with the reference was achieved by the MorphoDiTa Tagger (less than 95% for non-literary texts and less than 91% for literary texts). The highest performance was achieved by UDPipe2 (more than 99%), but there is no statistically significant difference between UDPipe2 and TreeTagger/RNNTagger/ Stanza ($p > 0.05$).

When determining the eleventh position, UDPipe2 appears to be the most powerful tool, but there is no statistical difference between it and the other automatic tools (RNNTagger, TreeTagger, and Stanza) in determining negation with respect to the reference, regardless of text type.

### The 12th tagset position—voice

The twelfth tagset position is only used for verb forms, mainly for verb participles (*Mikulová et al., 2020*).

A similar situation, as obtained in determining tense or negation, can also be observed in the case of voice.

Ranking of taggers determining the 12th position for non-literary and literary texts (Table 11) has identified two homogenous groups between which a statistically significant difference was observed in terms of agreement with the reference in the 12th position ($p < 0.05$). The lowest performance was again achieved by the MorphoDiTa and MorphoDiTa_online Tagger (less than 54% for non-literary texts and less than 47% for literary texts). The highest performance was by UDPipe2, Stanza, TreeTagger, and RNNTagger (more than 98% for non-literary texts and more than 99% for literary texts), which form one homogenous group ($p > 0.05$).

The only difference which can be observed is the order of taggers (Table 11), *i.e.,* in the case of literary texts, the most accurate tagger with a reference is UDPipe2, and in the case of non-literary texts, it is the RNNTagger.

## DISCUSSION

In the case of non-literary texts, the highest degree of concordance among the examined taggers (*Kendall Coefficient of Concordance > 0.4*) was identified for the tags in 8th position (*Kendall Coefficient of Concordance = 0.59*), 9th position (*Kendall Coefficient of Concordance = 0.50*), and 12th position in the tagset (*Kendall Coefficient of Concordance = 0.44*). We identified two homogeneous groups with similar performance in terms of accuracy (Tables 7a, 8a and 11a).

One homogeneous group consisted of MorphoDiTa and MorphoDiTa_Online, which achieved the lowest accuracy in the automatic determination of person, tense, and voice. All three positions in the tagset focus on the verb (the person of verb forms, verb forms in the purely morphological sense, and verb participles) during linguistic annotation. Our results indicate that neither tool is suitable for linguistic analysis of Slovak non-literary texts. On the other hand, tools like RNNTagger, TreeTagger, Stanza, and UDPipe2, when tagging the non-literary texts, achieved a high accuracy with reference for determining person (8th position), tense (9th position), and voice (12th position). They were consistent when it came to analyzing verbs and their forms and persons.

When determining other tag positions within the 15 positional tagsets that represent—part of speech, a detailed part of speech, gender, number, case, degree of comparison, and negation—automatic taggers achieved different quality. In general, MorphoDiTa achieved the lowest accuracy, and a statistically significant difference between it and MorphoDiTa_Online was proven. RNNTagger appears to be the most effective automatic tool, especially when it comes to determining part of speech, a detailed part of speech, gender, number, case, person, degree of comparison, and voice, even though when determining some tag positions (12th, 8th, or 4th) this tool was comparable to other taggers (Stanza/UDPipe2/TreeTagger).

Similarly, in the case of literary texts; the highest degree of concordance among the examined taggers (*Kendall Coefficient of Concordance > 0.4*) was identified for the tags in 8th position (*Kendall Coefficient of Concordance = 0.67*), 9th position (*Kendall Coefficient of Concordance = 0.86*), and 12th (*Kendall Coefficient of Concordance = 0.52*). We also identified two homogeneous groups with similar performance in terms to accuracy

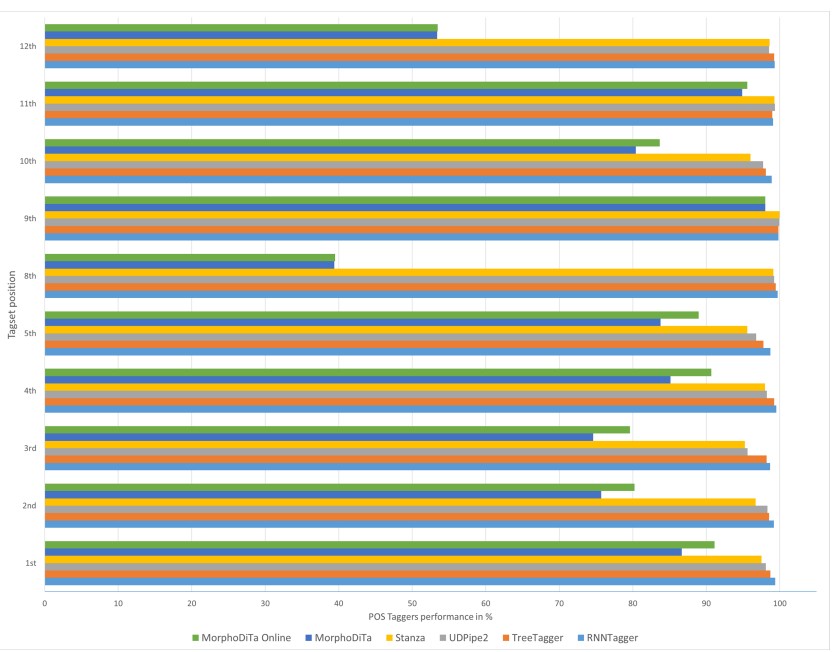

**Figure 2** Performance of automatic POS taggers on non-literary texts.

(Tables 7b, 8b and 11b), apart from the tag in 8th position (person) produced by the TreeTagger (Table 7b) and for the tag in 9th position (Table 8b) produced by the Stanza tagger (Table 11b).

When determining verbs, the same accuracy and performance of automatic taggers as for non-literary texts can be observed. However, when determining the other tags within the tagset, the most accurate determination is produced by UDPipe2, whether it is part of speech, a detailed part of speech, gender, case, degree of comparison, negation, and voice.

The results showed that the usage of automatic annotation tools could be proficient in the case of the Slovak language (Figs. 2 and 3). Four of the six examined tools achieved a high performance for most of the tagset positions. TreeTagger, as the predecessor of RNNTagger, lacked in some tagset positions (1st, 2nd, 3rd, 5th, 8th). The difference in performance was not large, but the new tool RNNTagger offers a novel method using recurrent neural networks for annotating texts.

Similar results were achieved for both text types (Figs. 2 and 3), and it can be concluded that usage of the RNNTagger should be preferred for both types. Stanza, as another representative of the neural network pipeline that is used for tagging, achieved high performance in almost all tagset positions (>95%). UDPipe2 also achieved a high performance, but mostly in the case of literary texts. In seven out of nine examined tagset positions of literary texts, UDPipe2 achieved the highest performance. In the case of non-literary texts, the highest performance was achieved by the RNNTagger (eight out of nine examined tagset positions). Together with RNNTagger, UDPipe2 achieved the highest performance of the examined taggers.

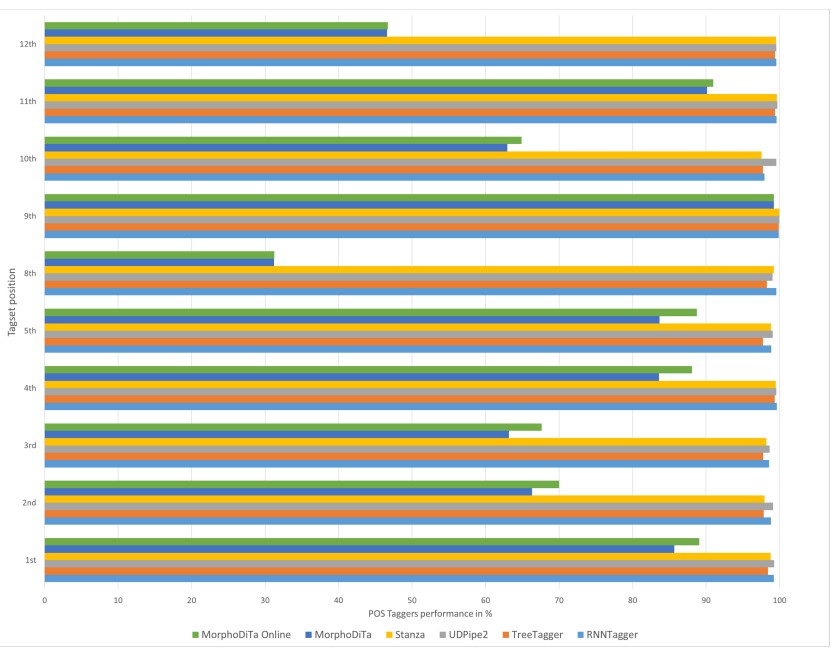

**Figure 3   Performance of automatic POS taggers on literary texts.**

On the other hand, MorphoDiTa and MorphoDiTa_online struggled in some tagset positions (3rd, 8th, 10th, 12th, 15th). The tool was designed by the same authors as UDPipe2 and was focused on the Czech language but supporting the Slovak language. The other issue that could have caused low performance could have been that they were the only tools that generated the output in the PST tagset format. As for the other tools a parser was used to convert the SNC tagset to PST tagset, it could have been that MorphoDiTa output was too detailed.

Our results indicate that it is useful to differentiate texts into literary and non-literary and subsequently, based on the text style to deploy a tagger. For literary text, UDPipe2 outperforms other taggers for morphological annotation of the inflectional Slovak language. However, for non-literary texts, RNNTagger is more effective (accurate) for morphological analysis compared to other taggers (Table 12). Moreover, our results show how linguistic aspects affect tagger performance in terms of accuracy with gold tokenization. For example, if we focus on number or person phenomena within the literary texts, it is more effective to deploy an RNNTagger; or in the case of morphological analysis focusing on tense, it is better to deploy Stanza, despite the fact that, in general, UDPipe2 performs the best.

Last but not least, our results reveal that the most effective approach to morphological annotation involves a combination of UDPipe2 and RNNTagger for general (non-specific) linguistic analysis.

Benko et al. (2024), *PeerJ Comput. Sci.*, DOI 10.7717/peerj-cs.2026

**Table 12  Ranking of taggers accuracy according to aspect of linguistic analysis.**

| Tag position | Literary texts | | | | | | Non-literary texts | | | | | |
|---|---|---|---|---|---|---|---|---|---|---|---|---|
| | MorphoDiTa | MorphoDiTa_Online | UDPipe2 | Stanza | TreeTagger | RNNTagger | MorphoDiTa | MorphoDiTa_Online | UDPipe2 | Stanza | TreeTagger | RNNTagger |
| Part of speech | | | **x** | | | x | | | | | | **x** |
| A detailed part of speech | | | **x** | | | x | | | | | | **x** |
| Gender | | | **x** | x | | x | | | | | | **x** |
| Number | | | x | x | x | **x** | | | | | x | **x** |
| Case | | | **x** | x | | x | | | | | | **x** |
| Person | | | x | x | | **x** | | | x | x | x | **x** |
| Tense | | | x | **x** | x | x | | | x | **x** | x | x |
| Degree of comparison | | | **x** | | | | | | | | | **x** |
| Negation | | | **x** | x | x | x | | | x | x | x | **x** |
| Voice | | | **x** | x | x | x | | | x | x | x | **x** |

**Notes.**

marked—the best performance in accuracy for individual text styles.

# CONCLUSIONS

The research was focused on evaluation the tagging functionality of various automatic tools for the Slovak language. Morphological annotation is a time-consuming task that requires lot of manual work from experts. The six analyzed automatic tools were evaluated based on the performance of the taggers expressed in terms of accuracy with gold tokenization. The results showed that all tools offer a high performance in determining the part of speech. That is important and offers a good baseline to use the tools. A more accurate complex morphological annotation of the word POS tag offered mainly RNNTagger and UDPipe2. Non-literary texts offered various genres, and RNNTagger achieved the highest performance in terms of agreement with the gold tokenization. Literary texts comprised novels and fairy tales where UDPipe2 achieved the highest performance in terms of agreement with the gold tokenization. High performance results were achieved also for TreeTagger and Stanza taggers on both text types.

The study has certain limitations, which mainly consist of the size of the dataset. This is an issue that is hard to resolve as creating a corpus with manual annotation is very time-consuming and requires a great deal of manual work. That is also the reason so many automatic annotation tools have been developed. Despite that, the used dataset was sufficient to highlight that many tools already support an inflectional language such as Slovak. This article also focused only on evaluating the tag generation performance within the 15 positional tagsets for the Slovak language. In future work, it would be appropriate to focus on lemmatization, as most of these tools also offer this functionality.

## Funding

This work was supported by the Scientific Grant Agency of the Ministry of Education of the Slovak Republic (ME SR), by the Slovak Academy of Sciences (SAS) under contract No. 1/0734/24, and by the Slovak Research and Development Agency under contract No. APVV-18-0473. The funders had no role in study design, data collection and analysis, decision to publish, or preparation of the manuscript.

## Grant Disclosures

The following grant information was disclosed by the authors:
Scientific Grant Agency of the Ministry of Education of the Slovak Republic (ME SR).
the Slovak Academy of Sciences (SAS): 1/0734/24.
the Slovak Research and Development Agency: APVV-18-0473.

## Competing Interests

The authors declare there are no competing interests.

## Author Contributions

- Lubomír Benko conceived and designed the experiments, performed the experiments, analyzed the data, prepared figures and/or tables, authored or reviewed drafts of the article, and approved the final draft.

- Dasa Munkova conceived and designed the experiments, prepared figures and/or tables, authored or reviewed drafts of the article, and approved the final draft.
- Mária Pappová conceived and designed the experiments, performed the experiments, performed the computation work, prepared figures and/or tables, and approved the final draft.
- Michal Munk analyzed the data, prepared figures and/or tables, and approved the final draft.

## Data Availability

The raw words with specific morphological tags in a dataset and the code are available in the Supplementary Files.

## Supplemental Information

Supplemental information for this article can be found online at http://dx.doi.org/10.7717/peerj-cs.2026#supplemental-information.

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
