# Peer review of "Comparison of various approaches to tagging for the inflectional Slovak language"

_PeerJ Computer Science, doi:10.7717/peerj-cs.2026_

## Round 0.1 · original submission · Major Revisions

Based on the reviewers’ comments, you may resubmit the revised manuscript for further consideration. Please consider the reviewers’ comments carefully and submit a list of responses to the comments along with the revised manuscript.

**Language Note:** PeerJ staff have identified that the English language needs to be improved. When you prepare your next revision, please either (i) have a colleague who is proficient in English and familiar with the subject matter review your manuscript, or (ii) contact a professional editing service to review your manuscript. PeerJ can provide language editing services - you can contact us at [email protected] for pricing (be sure to provide your manuscript number and title). – PeerJ Staff

Reviewer 1 ·

Basic reporting

The authors in this paper titled “Comparison of various approaches to tagging for the inflectional Slovak language” attempted to compare the six different automatic taggers for inflectional Slovak language. The authors reported interested findings, however there are several week points in the paper that requires further attention:
1: In abstract “….language, seeking the best-performing tagger in terms of accuracy with the reference.” What is reference means in this sentence? It is suggested that this word should be replace by proper wording such as “existing literature”, “literature”, or “benchmark”. Similarly, do the suggested wording change for the whole manuscript.
2: The summary to key results presented in abstract is a bit ambiguous, like “….both text types….”. The types are not defined in abstract, what both refer to. It is suggested to rewrite the abstract in a more structured format.
3: Highlight the limitations of existing research gaps this study is addressing along with research contributions being made at the end of introduction section.

Experimental design

4: The structure and organization of the manuscript at the end of introduction section should be improved if the sections are referred by name rather than next, second, third etc. section.
5: POS Taggers section is suggested to be placed after related work or as a part of methodology. Currently this section is hanging between introduction and related work section.
6: The section research methodology should be arranged based on the steps in figures. The figures needed to be redrawn with more details. Restructure whole section and describe each step-in detail along with evaluation metrics being used in this study.

Validity of the findings

6: Where possible compare the results with existing studies.

Cite this review as

Reviewer 2 ·

Basic reporting

The paper under review compared six different tagging approaches for the inflectional Slovak language. The work presented in the paper is aligned with the scope of the journal. My primary concern is that the authors evaluated existing approaches without presenting any of their approach. Considering this fact, this should be a review paper rather than a regular research paper. Moreover, the authors did not provide any justification of the parameters they have used to compare these six existing approaches.

Experimental design

Presently the presentation of the experimental results is not very affective. The authors included ten different tables in order to compare the tagging at ten different positions. It would be better to consolidate these results into less number of tables. They should present a side-by-side comparisons of all six approaches. The discussion section should present a learned lesson to the readers. Presently it is missing. Also, the dataset on which the experiments have been carried out is clearly defined. The authors should include the hardware specification used to run these experiments.

Validity of the findings

The figures are fine but need to clearly explain in the text. The tables have to be consolidated.

Additional comments

I'd suggest the authors to add a motivation section in the paper. Also, the contributions should be highlighted in bullet points form in the introduction section.

Cite this review as

---

## Round 0.2 · Minor Revisions

Based on the reviewers’ comments, you may resubmit the revised manuscript for further consideration. Please consider the reviewers’ comments carefully and submit a list of responses to the comments along with the revised manuscript.

Reviewer 1 ·

Basic reporting

The comments are addressed properly. However, currently there is only one figure in the whole manuscript and its quality is also not good. The authors are suggested to redraw methodology figure with more detailed components and better visibility rather than just showing the flow.
In addition, also add charts and visualization for the results section which will be more helpful in understanding the results.
The authors are also suggested to add a table to provides a overview/summary of existing literature at the end of related work section.

Experimental design

no comment

Validity of the findings

no comment

Additional comments

no comment

Cite this review as

Reviewer 2 ·

Basic reporting

The authors have addressed my all comments. No more comments from my side.

Experimental design

No more comments from my side.

Validity of the findings

No more comments from my side.

Cite this review as

---

## Round 0.3 · accepted · Accept

Congratulations, the reviewer is satisfied with the revision and recommended an accept decision.

Reviewer 1 ·

Basic reporting

no comment

Experimental design

no comment

Validity of the findings

no comment

Cite this review as